# The Sin3A/MAD1 Complex, through Its PAH2 Domain, Acts as a Second Repressor of Retinoic Acid Receptor Beta Expression in Breast Cancer Cells

**DOI:** 10.3390/cells11071179

**Published:** 2022-03-31

**Authors:** Nisha Rani Dahiya, Boris A. Leibovitch, Rama Kadamb, Nidhi Bansal, Samuel Waxman

**Affiliations:** 1The Tisch Cancer Institute, Icahn School of Medicine at Mount Sinai, New York, NY 10029, USA; nisha.rani@mssm.edu (N.R.D.); aachi22@gmail.com (N.B.); 2Department of Pathology, New York University School of Medicine, New York, NY 10029, USA; boris.leibovitch@nyulangone.org; 3Department of Cell Biology, Albert Einstein College of Medicine, Bronx, NY 10461, USA; rama.kadamb@einsteinmed.edu

**Keywords:** breast cancer, E-box, HDAC: histone deacetylases, MAD1: MAX dimerization protein, NCoR: nuclear receptor corepressor 1, PAH: paired amphipathic domains, RARβ: retinoic acid receptor beta, RARE: retinoic acid response element, Sin3A, co-immunoprecipitation

## Abstract

Retinoids are essential in balancing proliferation, differentiation and apoptosis, and they exert their effects through retinoic acid receptors (RARs) and retinoid X receptors (RXRs). RARβ is a tumor-suppressor gene silenced by epigenetic mechanisms such as DNA methylation in breast, cervical and non-small cell lung cancers. An increased expression of RARβ has been associated with improved breast cancer-specific survival. The PAH2 domain of the scaffold protein SIN3A interacts with the specific *S*in3 Interaction *D*omain (SID) of several transcription factors, such as MAD1, bringing chromatin-modifying proteins such as histone deacetylases, and it targets chromatin for specific modifications. Previously, we have established that blocking the PAH2-mediated Sin3A interaction with SID-containing proteins using SID peptides or small molecule inhibitors (SMI) increased RARβ expression and induced retinoic acid metabolism in breast cancer cells, both in in vitro and in vivo models. Here, we report studies designed to understand the mechanistic basis of RARβ induction and function. Using human breast cancer cells transfected with MAD1 SID or treated with the MAD SID peptide, we observed a dissociation of MAD1, RARα and RARβ from Sin3A in a coimmunoprecipitation assay. This was associated with increased RARα and RARβ expression and function by a luciferase assay, which was enhanced by the addition of AM580, a specific RARα agonist; EMSA showed that MAD1 binds to E-Box, similar to MYC, on the RARβ promoter, which showed a reduced enrichment of Sin3A and HDAC1 by ChIP and was required for the AM580-enhanced RARβ activation in MAD1/SID cells. These data suggest that the Sin3A/HDAC1/2 complex co-operates with the classical repressors in regulating RARβ expression. These data suggest that SIN3A/MAD1 acts as a second RARβ repressor and may be involved in fine-tuning retinoid sensitivity.

## 1. Introduction

Retinoic acid receptors (RARs) are non-hormonal receptors, classified as type II nuclear receptors, which reside in the nucleus, bound to their DNA response elements, called Retinoic acid response elements (RARE), even in the absence of a ligand [1]. Their intracellular ligand retinoic acid (RA) allosterically controls the interaction of these receptors with coactivators [2] or corepressors [3] by modulating the conformation of a short helix, called activation function 2 (AF2), at the C-terminal of the ligand-binding domain (LBD) of the receptor. In the absence of RA, the AF2 is in an open conformation, and the RARs are in a complex with histone deacetylases (HDAC) containing corepressors NCoR and SMRT. Upon the binding of RA to the ligand-binding domain (LBD), the AF2 forms one side of a charge clamp which grips the LxxLL helix end of coactivator proteins [3] associated with histone acetyltransferases (HATs). This facilitates formation of an open and activated conformation of chromatin.

Retinoids exert their diverse biological effects, such as the regulation of development, differentiation, cell proliferation and apoptosis, through RARs [4]. Retinoids have been studied for their chemo-preventive effect in solid tumors such as head and neck cancers [5], hepatocellular carcinomas, lung cancer [6], breast cancer and differentiation therapy for acute promyelocytic leukemia (APL) [7,8,9]. The significance of RAR activity in regulating the development of mammary adenocarcinomas was highlighted by a study by Kapumbati et al., where they showed that transplanting mammary epithelium from transgenic mice carrying a dominant negative RARα (RARαG303E) into the cleared fat pads of wild-type FVB mice resulted in the development of a metastatic mammary adenocarcinoma in one of the four transplanted glands 17 months post-transplantation [9]. Using the same model, the authors showed that retinoic acid receptors in adipocytes affect mammary morphogenesis through the paracrine loop [10]. Transcriptional activity directed by the retinoic X receptor and various nuclear receptors, including the estrogen receptor, progesterone receptor and vitamin D receptor, is influenced by epigenetic mechanisms such as DNA methylation at CpG islands and histone modifications [11].

One of the effector genes of RAR and RXR is Retinoic acid receptor beta (RARβ), which is a tumor-suppressor gene [12], and its methylation has been associated with breast [13], cervical [14,15] and Non-Small Cell Lung Cancer [16]. In addition to epigenetic mechanisms, genetic mutations have been found to contribute to the risk of breast cancer. Studies have shown that breast cancer specimens reportedly showed a loss of heterozygosity on chromosome 3 at the 3p24 locus, a region which codes for RARβ along with other genes [17]. Such a specimen showed that the adjacent tissue with regular morphology exhibited a normal expression of RARα and RXRα, while it showed a loss of only RARβ. An increased expression of RARβ has been associated with improved breast cancer-specific survival [18]; therefore, interventions which can improve RARβ expression in solid cancers are highly desirable.

Sin3A (the dominant paralog of Sin3 in breast cancer cells [19]) is a chromatin modulator which acts as a molecular scaffold for the assembly of chromatin-modifier enzymes such as HDACs, and it interacts with the Sin3 Interaction domain (SID) of transcription factors through its first and second *p*aired *a*mphipathic *d*omains (PAH), PAH1 and PAH2, respectively. The transcription factor interaction can be targeted using SID decoys such as the SID peptide or small molecule inhibitors (SMI). In our published work, we have established that the treatment of triple-negative breast cancer (TNBC) cell lines with SMI MS-905, which targets the interaction between the MAD1 (MXD1) and PAH2 domain of Sin3A, exhibits remarkable phenotypes both in vitro and in vivo [20]. The SID decoy treatment induced epigenetic reprogramming and inhibited the growth of TNBC cells. The SMI MS-905 treatment activated Retinoic Acid Receptor alpha (RARα), reactivated RARα/β pathways and increased endogenous retinoic acid levels and synthetic RARE-driven promoter activity [20]. In a BALB/c xenograft model, the SMI showed remarkable synergy with the RARα agonist AM80, which was superior to an ATRA (all-trans retinoic acid)/MS-905 combination under a post-surgical adjuvant setting. The AM80/MS-905 combination showed 100% disease-free survival and inhibition of lung metastasis, which is highly significant from a clinical perspective. These data suggest that the SID decoy treatment enhances the efficacy of retinoids, indicating an association between nuclear receptor corepressor complexes and the transcription factor-bound Sin3/HDAC complex. We have also shown that MDA-MB-231 cells treated with an MAD1-SID decoy peptide showed a modest increase in global acetylation levels of histone H3 and significant increases in H3K4 methylation, particularly for CDH1 and RARβ genes [21], indicating the activation of these genes. Additionally, evidence emanating from studies on ATRA differentiation-resistant APL suggests that a functionally active RARα receptor is essential for the effectiveness of HDAC inhibitors such as sodium butyrate (NaB) [22].

Rosenfeld and his coworkers described that the combination of two repressor complexes such as NCoR/HDAC3 and Sin3/HDAC block expression of the target gene of the thyroid receptor [23]. Using the GAL4-UAS reporter system, they observed a strong activation of the reporter gene after removal of the Sin3/HDAC complex. Based on these data and our data for MAD1/Sin3/HDAC1/2, we speculate that both complexes repress the RARβ gene by carrying the HDAC cargo near the transcription start site of the RARβ gene and that in the presence of ligand, only NCoR gets displaced from the complex, relieving only partial repression. The addition of SID decoys leads to a release of the MAD1/Sin3A/HDAC1 repressor complex, resulting in further RARβ expression. In the present study, we have demonstrated the interplay among these corepressor complexes in regulating the expression of both the RARα and RARβ gene. This work also provides insights into the mechanism through which SID decoys activate retinoic acid signaling. Such information is essential for our understanding of generalizability for the use of SID decoys as a single agent or combination drug treatment with retinoids for TNBC cells.

## 2. Material and Methods

### 2.1. Cell Lines and Culture Media

Human mammary adenocarcinoma cell lines, Michigan Cancer Foundation-7 (MCF-7, a luminal phenotype) (Cat# HTB-22) and triple negative M.D. Anderson Breast-231 (MDA-MB-231) (Cat# HTB-26) cells and the mouse adenocarcinoma cell line 4T1 (Cat# CRL-2539) were purchased from American Type Cell Culture. The MCF-7 and MDA-MB-231 cell lines were maintained in DMEM supplemented with 10% fetal bovine serum and 1% antibiotic–antimycotic solution (Invitrogen), while 4T1 was maintained in a DMEM/F12 medium supplemented with 5% FBS and 1% antimycotic–antibiotic solution. The cell lines were authenticated in February 2019 (Genetica Cell Lines, cases CX4006499 and CX4006499) by short tandem repeat (STR) profiling in accordance with the standard ASN-0002-2011.

### 2.2. Synthetic Peptide Decoys and Generation of SID-Transfected Stable Cell Lines

The MAD1 (also known as MXD1) SID peptide (sequence: YGRKKRRQGGG-VRMNIQMLLEAADYLERRER), referred to as MAD-SID throughout the manuscript, and its scrambled version, Scr peptide (MAD-Scr) (sequence: YGRKKRRQGGGEQRARRIMERLLEYNMVADL), were synthesized to a purity level of 95% (BioSynthesis, Inc., Lewisville, TX, USA) and were used in all the experiments at a concentration of 5 µM for the 24 h treatment and 2.5 µM for the 72 h treatments. In order to generate stable cell lines overexpressing MAD1-SID sequences, MCF-7 and MDA-MB-231 cell lines were transfected with a SID sequence containing vector (MAD-SID), or empty vector (pCMV) as control, using the turbofect transfection reagent. The transfected cells, expressing a GFP tag, were FACS sorted and maintained for further use.

### 2.3. Protein Extraction and Western Blotting

Stably transfected MCF-7 and MDA-MB-231 cells or peptide-treated wild type MCF-7 and MDA-MB-231 cells were collected and lysed using an RIPA buffer. The BCA (Pierce kit #23225) method was used for protein quantitation of the whole cell lysates. The whole cell lysates were subjected to SDS-PAGE and Western blotting. Specific proteins were probed using anti-Sin3A (cat#sc-5299, Santa Cruz Biotechnology, Dallas, TX, USA), anti-RARα (cat# sc-515796), anti-RXRα (cat#sc-515929, Santa Cruz Biotechnology, Dallas, TX, USA) and anti-RARβ (cat# sc-56864 Santa Cruz Biotechnology, Dallas, TX, USA), followed by incubation with HR0 conjugated anti-mouse antibodies (cat# 7076, Cell Signaling technologies, Denvers, MA, USA). The bands were visualized by chemiluminescence.

### 2.4. Co-Immunoprecipitation

MDA-MB-231 and MCF-7 cells, either as wild type, or transfected with pCMV/MAD-SID, or treated with 5 µM of the MAD-Scr or MAD-SID peptide, were used for the coimmunoprecipitation assays. Treated cells were collected and lysed in a Pierce IP reagent, and 2000 µg of lysates were immunoprecipitated overnight with a primary antibody. The immune complex was immobilized on A/G beads, eluted in a 2X sample buffer and subjected to SDS-PAGE, followed by Western blotting. The amount of Sin3A bound to RARα/RXRα was visualized by chemiluminescence. Mouse IgG were used as a negative control, and a 10% input was used as internal control. For all the blots, including Western blots, the protein bands were quantitated using the ImageJ software. The quantitations of proteins for Western blots were normalized against β-actin, while the coimmunoprecipitated protein (Sin3A) was normalized against inputs (RARα and RXRα).

### 2.5. Luciferase Assays

To test the responsiveness of the RARβ promoter, three different clones encompassing the core promoter (endogenous/natural promoter) region of RARβ gene were generated, which were tagged with a luciferase sequence. The first clone, clone #30, encompassed the RARE sequence (GGTTCACCGAAAGTTCA), which is present at 3 bp upstream of the TATA box (Figure 3B); clone#24 included both the RARE sequence and the proximal E-box, and clone#16/21 contained the entire sequence, encompassing the RARE, proximal E-box and a distal E-box. These clones contained one copy of luciferase driven by either of these endogenous promoter regions. The sequence of all the constructs was confirmed through sanger sequencing and compared with the RARβ sequence available in literature [13,24]. The Cignal RARE reporter assay kit (cat#336841, Qiagen, Hilden, Germany), in which the reporter construct contained four copies of the RARE sequence AGGTCACCAGGAGGTCA, was used as a positive control. The responsiveness of these clones was determined by transfecting 293T cells with respective clones and treating the transfected cells with AM580 (cat#A8843, Millipore Sigma, Burlington, MA, USA) at a 500 nM concentration for 24 h alone or in combination with the MAD-SID or MAD1-Scr peptides. The promoter activity was estimated using a Dual-Glo Luciferase Reporter Assay System (cat#E1910, Promega, Madison, WI, USA), and the cells were processed as per the manufacturer’s protocol.

### 2.6. RT-PCR

Total messenger RNA was extracted from the control and treated cells using the RNAeasy Mini Kit (Qiagen, Hilden, Germany), followed by cDNA synthesis by reverse transcription using the iScript DNA Synthesis Kit (Bio-Rad, Hercules, CA, USA), and RT-PCR was conducted using the SYBR Green PCR Master Mix (Applied Biosystems, Foster City, CA, USA) with specific primers for RARα, RARβ, HOXA5, CDH1 and SOX2 genes. RPL30 was used as an internal control.

## 3. Chromatin Immunoprecipitation (ChIP)

MDA-MB-231 cells were plated and treated with 1 μM AM580 (Sigma) for 24 h, followed by crosslinking using 1% formaldehyde. Cell nuclei were isolated using hypertonic buffer A (150 mM NaCl, 50 mM Tris-HCl, pH 8, 1% NP40, 1% sodium deoxycholate, 0.5% SDS and 2 mM EDTA), centrifuged, lysed in SDS lysis buffer (50 mM Tris, pH 8, 10 mM EDTA, 1% SDS) and then sonicated. The chromatin obtained from the nuclear fraction was pre-cleared with protein A/G beads (SantaCruz Biotech) and mouse/rabbit immunoglobulin G (IgG). Sin3A, HDAC1/2 and NcoR were immunoprecipitated overnight at 4 °C, and the immunoprecipitates were incubated with protein A/G beads for 2 h at 4 °C. The immunoprecipitates were washed and eluted with an elution buffer (100 mM Na_2_CO_3_ and 1% SDS) from agarose beads and incubated in 5 M NaCl at 65 °C overnight. DNA was isolated by digestion with proteinase K, treatment with RNAse A and purification using the QIAquick DNA purification kit (Qiagen, Valencia, CA, USA). Primers for the detection of the RARβ promoter-specific E-Box and RARE sequence were designed. E-box sense: 5′-CTT GCC TAC CCT GAT GGT GT-3′, and E-box-antisense: 5′-CCC TTC TCA CCT GCT ACC TG-3′; RARE sense 5′-CTT GCC TAC CCT GAT GGT GT-3′ and antisense 5′-CTT GCC TAC CCT GAT GGT GT-3′ were synthesized (Invitrogen). As a control, nonspecific primers were designed using a region 5000-bp upstream of the RARβ promoter, nonspecific-sense 5′-GCA AGA CTG CTT GCT CTC CT-3′ and nonspecific-antisense 5′-AAC ACA TCG TGG GTG GTC TT-3′.

### Electrophoretic Mobility Assay (EMSA)

One set each of double-stranded probe was designed, which included the proximal (5′AGGGCTTGCATGTGCTTTTTCTG) and distal E-box sequence (5′AATGTCTTAACAGCTGGCCATTGGTT), and it was synthesized and biotinylated. A non-radioactive complete EMSA kit for c-Myc from Viagene Biotech Inc. (BITF253) was used, and the supplier protocol was followed for gel preparation, standardizing gel running conditions and visualizing the EMSA shift. Probes supplied with the kit which included the consensus E-box sequence (5′AGCAGACCACGTGGTCTGCTTC) were used as positive control. Two micrograms of purified proteins c-Myc (abcam #ab169901) and MAD1 (abcam #ab82180) were incubated with the probes, run on 5% native polyacrylamide gel and transferred to a nylon membrane, followed by DNA crosslinking to the membrane and visualization as per the supplier’s protocol.

## 4. Results

### 4.1. MAD1 Regulates RARα through the SIN3A PAH2 Domain

We have previously reported that MAD-SID expression in triple negative breast cancer cells induced differentiation in 2-D cultures, morphogenesis in 3-D cultures and exerted anti-tumor effects in SID expressing MMTV-Myc cells derived from MMTV-c-Myc tumors injected into the fat pads of syngeneic friend virus B-type (FVB) mice [21]. In addition, we demonstrated that MAD-SID-expressing cells showed increased levels of H3K4 methylation in the RARβ promoter [21]. In another study, we observed that the treatment of MDA-MB-231 and 4T1 cells with a small molecule inhibitor SID decoy for 96 h resulted in increased expression of RARβ [20]. To study the mechanistic basis of increased retinoic acid receptor expression, we generated MDA-MB-231 and MCF-7 cells stably expressing the GFP-tagged MAD-SCR (referred to as pCMV) and MAD-SID peptide sequence (Figure 1E) (Appendix A), and we sorted the transfected cells to achieve a transgene-expressing enriched population. A quantitative PCR performed on these cells showed an increase in the expression levels of RARα and RARβ (Figure 1A,B). The increase in the expression of RARα was highly significant in MCF-7 (*p* = 0.006) and MDA-MB-231 (0.0013) cells expressing MAD-SID, with a moderate increase in the expression of RARβ. The increased expression correlated with cell confluence and was much greater at 120 h of growth (Appendix A). The MAD-SID expressing MDA-MB-231 and MCF-7 exhibited an increased expression of CDH1 (*p* = 0.05 and *p* = 0.005, respectively), and the expression of SOX2 (a stemness marker) was found to be significantly reduced in MCF-7 cells expressing MAD-SID (*p* = 0.02) (Appendix A). MAD-SID overexpression sensitized MDA-MB-231 cells to AM580 (an RARα-specific agonist) treatment as early as 24 h (Figure 1C,D), as there was a significant increase in the expression of both RARα (*p* = 0.004) and RARβ (*p* = 0.048) as early as 24 h. In pCMV cells, however, the expression of RARα reduced significantly after treatment with AM580 within 24 h. Similar findings have been reported previously that the treatment with various agonists (trans-RA, AM580, 9-cis-RA and TTNPB) resulted in degradation of RARα after 12 h of treatment [25], suggesting that transcriptional up-regulation of nuclear receptors by their ligands may be a feedback mechanism allowing sustained target-gene activation. The increase in RARβ also led to an increase in the expression levels of HOXA5 (Appendix A), which is an RARβ target gene [26].

### 4.2. The MAD-SID Peptide Disrupts RARα/RXRα Interaction with Sin3A

Literature suggests that the RARα and RXRα heterocomplex is complexed with NCoR in the absence of a ligand [27], and NCoR also interacts with Sin3A at its PAH1 domain [28], meaning that RARα and Sin3A may interact indirectly as part of a super-complex with NCoR. We carried out immunoprecipitation assays to assess whether, firstly, there exists a direct or indirect association between RARα, the RXRα heterodimer and Sin3A in breast cancer cell lines and, secondly, whether MAD1-SID treatment would disrupt this interaction. Indeed, immunoprecipitation assays using RARα and RXRα as bait confirmed that Sin3A co-precipitated with RARα and RXRα in MDA-MB-231, 4T1 and MCF-7 cells (Figure 2A). Treatment with the MAD-SID peptide significantly reduced both the RARα and RXRα association with Sin3A in MCF-7 cells, while in MDA-MB-231 cells, only the interaction with RXRα showed a reduction, which correlated with the ability of the MAD-SID peptide to disrupt the MAD and Sin3A association in both the cells lines. (Figure 2B). The CoIP assays performed in MAD-SID-overexpressing cells showed a dissociation between MAD and Sin3A in MDA-MB-231 cells, as well as in MCF-7 cells, indicating that the effective amount of MAD-SID needed to disrupt the MAD1 and Sin3A interaction is higher in MDA-MB-231 cells (Figure 2B,C).

### 4.3. Putative E-Box in RARβ Promoter Binds with MAD1

We conducted an in silico analysis of the RARβ promoter and found that one DR5 RARE element, GGTTCACCGAAAGTTCA, is present 3 base pairs (bps) upstream of the TATA box and in two E-box sequences, one proximal (about 500 bp upstream of the transcription start site) and one distal (about 5000 bps upstream), which are putative MYC/MAD1 binding sites. These E-box sequences, CATGTG (proximal) and CAGCTG (distal), are variations of the canonical (minimally defined as CANNTG) E-box sequence CACGTG. Therefore, we conducted an EMSA assay to ascertain if the pure c-Myc and MAD1 proteins would bind to these sequences (Figure 3A). Myc and MAD1 belong to the b-HLH-LZ family of transcription factors. Heterodimers of Myc/Max or MAD1/Max bind their cognate DNA sequence, known as the E-box (CACGTG). MAD1 and MYC can competitively bind to the same E-box in a context-dependent manner. An EMSA assay was performed by incubating purified c-myc and MAD1 proteins with 22 bp long respective probe sequences. Control probes supplied with the kit which contained the classic CACGTG were used as a control. The results demonstrate that c-Myc (lanes 2,5,8, Figure 3A) and MAD1 (lanes 3,6,9 Figure 3A) showed strong binding to the classic E-box sequence in lanes 2 and 3, respectively; the proximal E-box showed moderate binding, as shown in lane 5 and 6, respectively, while the distal E-box sequence showed very weak binding. The results show that MAD1 binds more strongly than c-Myc, which is contrary to the data available in literature. The differences in the binding intensity of c-Myc and MAD1 to various E-box sequences can be attributed to the quality of the purified proteins.

### 4.4. Members of SIN3A Complex Are Present at the RARβ Promoter

ChIP assays conducted on MDA-MB-231 cells showed the presence of Sin3A (*p* = 0.051), which was borderline significant, and MAD1 on the E-box of the RARβ promoter and RARα on the RARE region (Figure 3B,C). In MDA-MB-231 cells transfected with the MAD-SID transcript, the amount of Sin3A (*p* = 0.037) and HDAC1 (*p* = 0.053) reduced on the E-box of the promoter, which was found to be borderline significant (Figure 3D). Interestingly, the enrichment of MAD1 increased significantly (*p* = 0.005) on the E-box; however, this might not affect the activation of RARβ, as the repressive activity is related with Sin3A and HDAC1. NCoR/HDAC3 is the classic repressor complex associated with RAR/RXR. Rosenfeld et al. demonstrated that the N-terminal of NCoR interacts with the HID domain of Sin3A; therefore, it is conceivable that both might form a co-repressor super-complex. Our data suggest that the Sin3A/HDAC1/2 complex co-operates with the classical repressors in regulating RARβ expression. ChIP assays showed that NCoR is enriched at the RARE region and that there is no significant difference between the enrichment of NCoR in the MDA-MB-231 pCMV and MAD-SID carrying cells; however, enrichment was significantly reduced after AM580 treatment (Figure 5A), both in AM580-treated pCMV (*p* = 0.05) and MAD-SID-overexpressing cells (*p* = 0.051).

### 4.5. The RARβ Promoter Responds to AM580 and MAD-SID Treatment

In order to demonstrate that the two sequences (RARE and E-box) have functional relevance for the activation of the RARβ promoter, we conducted luciferase assays with various regions of the RARβ promoter driving luciferase expression. We generated three constructs, which encompassed three different regions of the native RARβ promoter. Results from the luciferase assays showed that all the clones responded to the AM580 treatment. The relative luciferase units (RLUs) of the AM580-treated wells were significantly higher than the RLUs of DMSO wells in RARE-containing clone#30 (*p* = 0.0008) and the positive control (*p* = 0.02), as well as in the E-box and RARE-containing clones 16/21 (*p* = 0.003) and clone 24 (0.0001). Clone#30, which contains the RARE sequence, responded similarly to the positive control, which was synthetic RARE. An enhanced luciferase signal was also observed after treatment with the MAD-SID peptide. In fact, the MAD-SID treatment of clone#24 (RARE and E-box) showed higher RLUs (*p* = 0.03) as compared to clone#30 or the positive control, which only contained the RARE sequence (Figure 4C), which shows that removal of the Sin3A complex from the RARβ promoter showed higher activation, which is in line with Rosenfeld’s findings. The results indicate that AM580 achieves higher activation of the RARβ promoter than the MAD-SID peptide; however, the peptide formulations may be less efficiently absorbed by the cells or degraded after uptake, which may affect its efficacy. Both the treatments, when given together, showed an additive effect in clone#24 as compared with AM580 (Figure 4D). The positive control, which was synthetic RARE, showed an increase in RLUs when compared with DMSO (Figure 4C); however, the combination of the SID peptide and AM580 did not show a significant increase in RLUs as compared with AM580 (Figure 4D).

## 5. Discussion

Retinoic acid receptors act as molecular switches [29] by acting in the absence of a ligand as promoter-bound transcription repressors associating with corepressors, and on the contrary, in the presence of ligand, they complex with coactivators to facilitate activation of transcription [30]. In our published work, we have demonstrated that targeting the interaction of Mad1 with the PAH2 domain of Sin3A using a small molecule inhibitor (SMI) confers retinoid sensitivity in triple negative breast cancer cells (TNBC) by upregulating the expression of RARα and RARβ and by increasing retinoic acid metabolism in TNBC [20].

We have also shown that MDA-MB-231 cells treated with a MAD-SID peptide showed a modest increase in histone H3 global acetylation and a significant increase in H3K4 methylation, particularly for the promoters of CDH1 and RARβ genes [21], indicating the activation of these genes. In the present study, we have dissected the molecular mechanisms involved in the activation of retinoic acid receptors and the role of targeting the PAH2 domain of Sin3A in regulating their expression. It is evident from the luciferase assay results that the MAD-SID treatment activated RARβ in the clone containing only the RARE sequence and the clone containing both the RARE and E-box sequence (Figure 4B). Importantly, the activation of the clone containing both the RARE and E-box sequence was higher, indicating the ability of the Mad-SID peptide to activate the RARβ promoter, even in the absence of an RARα-specific ligand (Figure 4B). The expression data also suggest that after AM580 treatment, MAD-SID-overexpressing cells show a rapid induction of RARβ (Figure 1D). This increase in the levels of RARα was also observed at the protein level, both in MDA-MB-231 cells and MCF-7 cells, while there was a minimal increase in the RARβ protein levels (Appendix A). The minimal increase in RARβ at the protein levels might be attributable to another level of post-transcriptional regulation of RARβ expression in TNBC, such as regulation by long non-coding RNAs. These data, taken together, indicate that the Sin3A-Mad-HDAC complex might be involved in the regulation of RARβ expression, and it also co-operates with RARα and its associated corepressor complex.

In our study, we showed that the Sin3A/HDAC complex participates in this repression. The EMSA and CHIP-PCR data showed that the E-box sequence near to the TSS site of RARβ binds with MAD, and this sequence showed the presence of Sin3A and HDAC1. The CHIP-PCR data also demonstrated that the expression of Mad-SID does not impact the interaction of NCoR with RARE; the interaction is only affected by AM580 treatment. On the other hand, the MAD-SID-expressing TNBC cells showed reduced recruitment of Sin3A and HDAC on the E-box of RARβ, which was borderline significant (Figure 3D). The increase in RARα expression in the MAD-SID-overexpressing cells is likely to be regulated by a similar mechanism, as a search of the Integrative Genomics Viewer (IGV) database for putative MAD1 binding sites yields multiple E-boxes in the RARα promoter (data not shown). The RARα/RXR heterocomplex and NCoR complex are known repressors associated with RARα activation; therefore, it is valid to anticipate that a similar mechanism regulates RARα expression. The preliminary CHIP data for some E-box-containing regions were inconclusive; however, they necessitate further experiments.

Similar studies have been published where NcoR and non-liganded RARα have been shown to form a complex to repress hormone-responsive genes [31]. In a study by Nagy et al., Sin3A was shown to form a multi-subunit repressor complex with SMRT and HDAC1, and it was shown that HDAC inhibitors synergize with retinoic acid and induce differentiation in myeloid leukemia (HL-60) cells [32]. In a retinoid-specific context, retinoid 6-[3-(1-adamantyl)-4-hydroxyphenyl]-2-naphthalenecarboxylic acid binds to a nuclear receptor called the small heterodimer partner (SHP) and recruits a complex containing Sin3A, N-CoR, HDAC4 and HSP90 [33].

There are further instances where Sin3A has been reportedly associated with other corepressors. Moehren et al. have demonstrated that Sin3A and Alien associate with the vitamin D3 receptor and are found on the D3 response elements of the human 24-hydroxylase (CYP24) promoter, and it enhances Alien-mediated gene repression [34]. Sin3A interacts with a number of transcription factors through its second paired amphipathic α-helix domain (PAH2) [35,36,37,38,39], which includes MAD1. Mad is a dedicated repressor which complexes with Sin3A constitutively to carry out gene repression. Nagy et al., in their paper, have suggested that since Sin3A is associated with both NCoR and MAD1, they proved a convergence in the repression pathways associated with bHLH-Zip proteins and nuclear receptors [32] by demonstrating that HDAC inhibitors and retinoic acid synergistically stimulated hormone responsive genes and the differentiation of myeloid leukemia cells, which has been further confirmed by the present study. Similarly, SMI to PAH2 increased AM580’s anti-tumor efficacy, decreased metastases and improved survival in in vivo TNBC mouse models, supporting the importance of the removal of the Sin3A repressor complex [20]. Thus, partially repressed RARα/β expression may explain the minimal clinical activity using retinoids such as ATRA as a single agent in breast cancer treatment.

Another implication of targeting the Sin3A and MAD1 interaction would be to destabilize the putative Sin3A-HDAC1/NCoR-HDAC3. Structural studies have shown that the domains of a protein/complex mutually participate in stabilizing each other’s interactions with external binding proteins, such as transcription factors. Nagy et al. demonstrated that the PAH1 domain of mSin3A interacts with SMRT; however, the inclusion of PAH2 strengthens the interaction and association between Sin3A, and SMRT is strongest when all of the four domains are included [32]. In another example, it was shown through molecular modeling that PAH1-bound sulfatide in Sin3B destabilized the Mad1–PAH2 interaction by affecting the stability of the α helices of PAH2 [40]. Our results from Co-IP, luciferase and CHIP experiments indicate that even if Sin3A and NCoR corepressors are forming a complex, destabilization of the internal interactions of Sin3A does not affect the association of NCoR with RARE. However, the removal of NCoR might affect the association of Sin3A with the E-box, which was shown by the Co-IP experiments.

Our work is indicative of a co-operation between NCoR and Sin3A in repressing RARβ expression. Our data show that in addition to the RARα and NCoR complex, the MAD-associated Sin3A/HDAC complex might be mediating the repression of RARβ. The introduction of an RARα-specific ligand displaced NCoR from the nuclear receptor, achieving partial de-repression. However, when we introduced PAH2-directed decoys, it also removed the second co-repressor, leading to complete dissociation of the repressor complex, causing further de-repression of the RARβ gene (Figure 5B). Our data suggest that SIN3A/Mad1 acts as a second ligand-independent RARβ repressor and may be involved in fine-tuning retinoid sensitivity. Further experimental data are needed to confirm this hypothesis.

## Figures and Tables

**Figure 1 cells-11-01179-f001:**
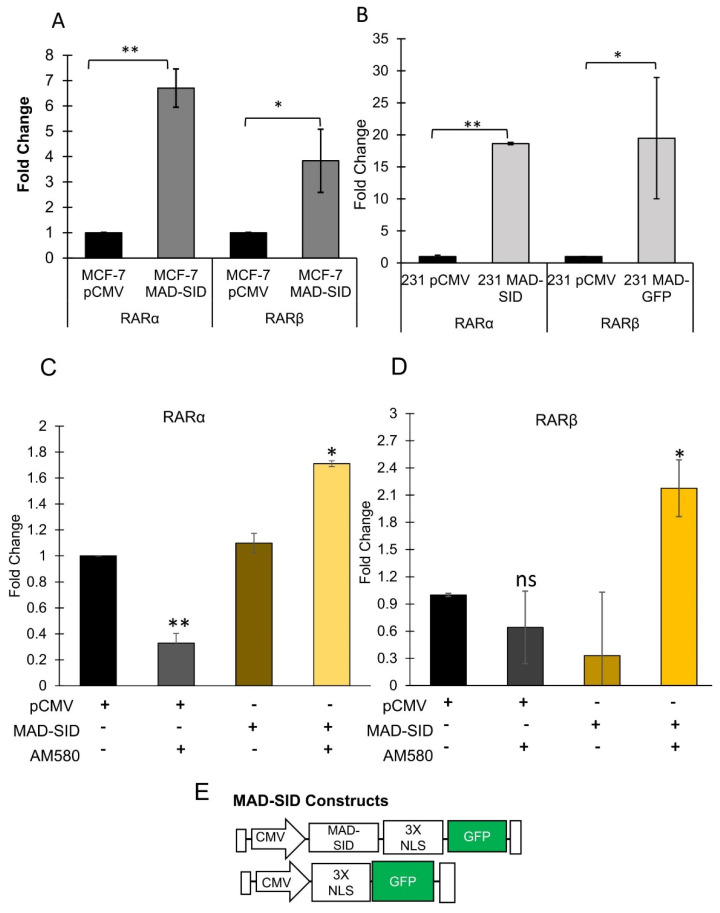
Expression of MAD-SID in MDA-MB 231 and MCF-7 cells increases RARα and RARβ Expression: (**A**,**B**) MDA-MB 231 and MCF-7 cells were transfected with CMV-driven plasmids carrying a nuclear localization signal, GFP tag and MAD-SID sequence. The transfected cells were examined for the expression of RARα and RARβ through RT-PCR. The expression of RARα and RARβ increased in both the cell lines. The data presented are the mean fold change obtained from four independent experiments (**C**,**D**). The expression of RARα and RARβ, respectively, was evaluated in the control (pCMV) and the MAD-SID-overexpressing MDA-MB-231 cells after treatment with AM580. The data presented are the average of the fold changes in expression obtained from two independent experiments. (**E**) Layout of the pCMV and MAD-SID constructs. The statistical analysis was conducted using an unpaired ‘*t*’ test; error bars represent mean ± SD. ** *p* < 0.01 and * *p* < 0.05.

**Figure 2 cells-11-01179-f002:**
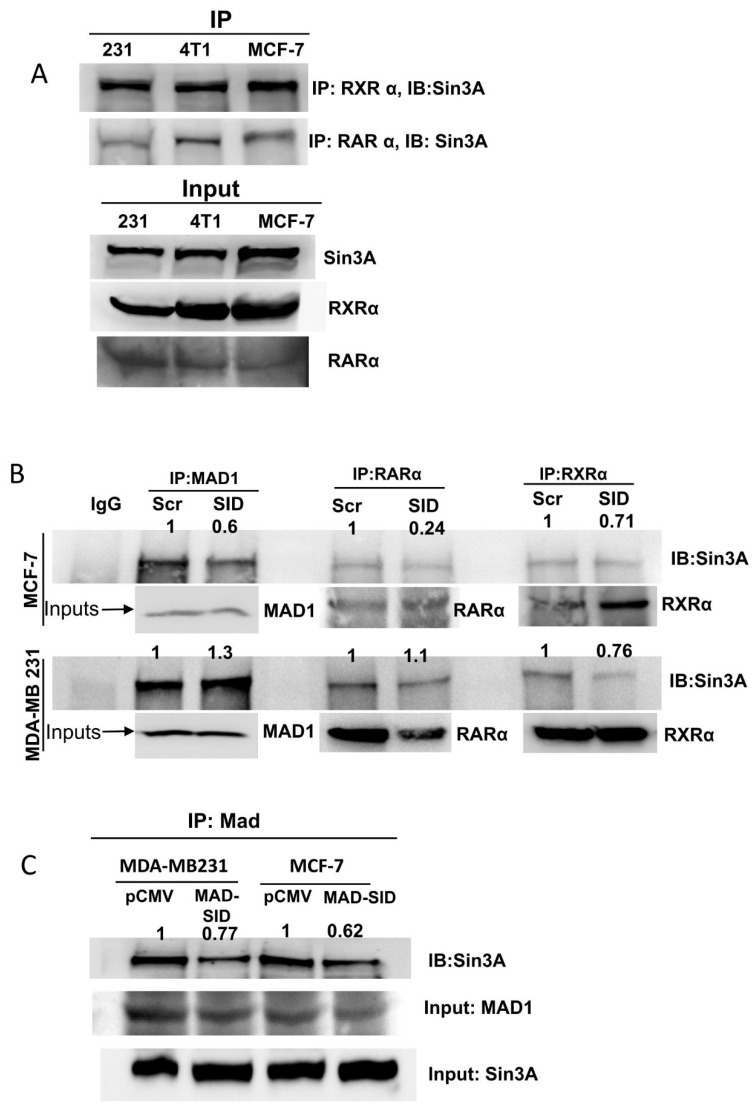
The MAD-SID peptide disrupts the RARα/RXRα interaction with Sin3A: Co-precipitation experiments were performed to verify the association between RARα, RXRα and Sin3A in breast cancer cells lines. Lysates of wild-type MDA-MB-231, 4T1 and MCF-7 were precipitated with RARα and RXRα antibodies, and the complex was precipitated on proteinA/G beads, eluted and subjected to SDS-PAGE and Western blotting. (**A**) A co-immunoprecipitation assay showed that Sin3A co-precipitated with both RARα and RXRα in all of the three cell lines. (**B**) The treatment of MCF-7 and MDA-MB-231 cells with 5 µM of the MAD-SID peptide for 24 h showed a reduction in RARα and Sin3A association in MCF-7 cells and of RXRα and Sin3A in both of the cell lines. (**C**) The Sin3A and MAD association was disrupted in MAD-SID-overexpressing MCF-7 and MDA-MB-231 cells. The peptide concentration was sufficient to disrupt the Sin3A–MAD interaction in MCF-7 cells but not in MDA-MB-231 cells. The dissociation in the Sin3A–MAD interaction was observed in MAD-SID-overexpressing MDA-MB-231 cells, indicating that a higher concentration of MAD-SID is needed for the disruption of interaction in these cells. The data presented are representative data obtained from three independent biological replicates.

**Figure 3 cells-11-01179-f003:**
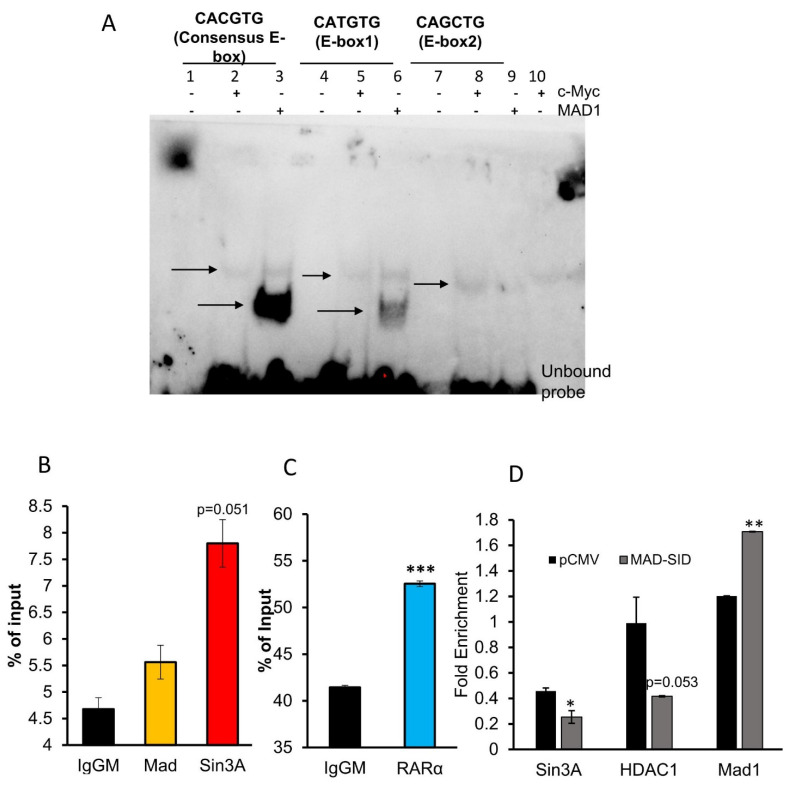
MAD-Sin3A Complex members are present at the RARβ promoter: (**A**) The Electrophoretic Mobility Shift Assay showed that Mad binds to the proximal E-Box of the RARβ promoter. Arrows indicate the shift in the movement of probes resulting from c-Myc (lanes 2,5,8) or MAD1-binding (lanes 3,6,9). The blot presented is a representative figure of four independent experiments. (**B**,**C**) The Chromatin immunoprecipitation Assay performed on the RARβ promoter of MDA-MB 231 cells showed the enrichment of Sin3A and MAD1 on the E-box and RARα on the RARE sequence of the RARβ promoter. The data presented are an average of three independent experiments. (**D**) The MAD-SID-overexpressing cells showed a reduced enrichment of Sin3A and HDAC1 on the E-box sequence of the RARβ promoter. The data presented are an average of two experiments. The statistical analysis was conducted using an unpaired ‘*t*’ test; error bars represent mean ± SD. *** *p* < 0.001, ** *p* < 0.01 and * *p* ≤ 0.05.

**Figure 4 cells-11-01179-f004:**
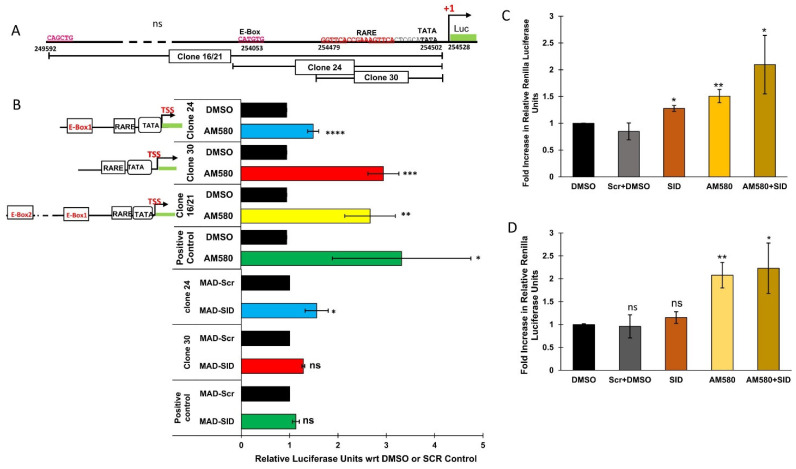
The RARβ promoter is controlled by both MAD1 and RARα: the RARβ promoter was analyzed for the presence of the E-Box and Retinoic Acid Response Elements (RARE). Luciferase clones driven by different regions of the RARβ promoter were generated. (**A**) Arrangement of the E-Box sequence about 500 bp upstream of the transcription start site and RARE about 5 bp upstream of the TATA box of the RARβ gene. The figure shows the layout of various clones generated to test the responsiveness of the RARβ promoter to different treatments. (**B**) Response of RARβ clones to AM580 and MAD-SID peptide treatment. Data presented are an average of the fold change in the renilla luciferase units from at least three independent experiments. (**C**) Response of RARβ clones to the combined treatment of AM580 and the MAD-SID peptide in an RARβ endogenous construct containing both the E-box and RARE (clone#24) to various treatments. (**D**) Response of synthetic RARE consisting of five DR5 sequences to various treatments. Data presented are an average of the change in the renilla luciferase units from three independent experiments. The statistical analysis was conducted using an unpaired ‘*t*’ test; error bars represent mean ± SD. **** *p* ≤ 0.0001, *** *p* < 0.001, ** *p* < 0.01 and * *p* < 0.05.

**Figure 5 cells-11-01179-f005:**
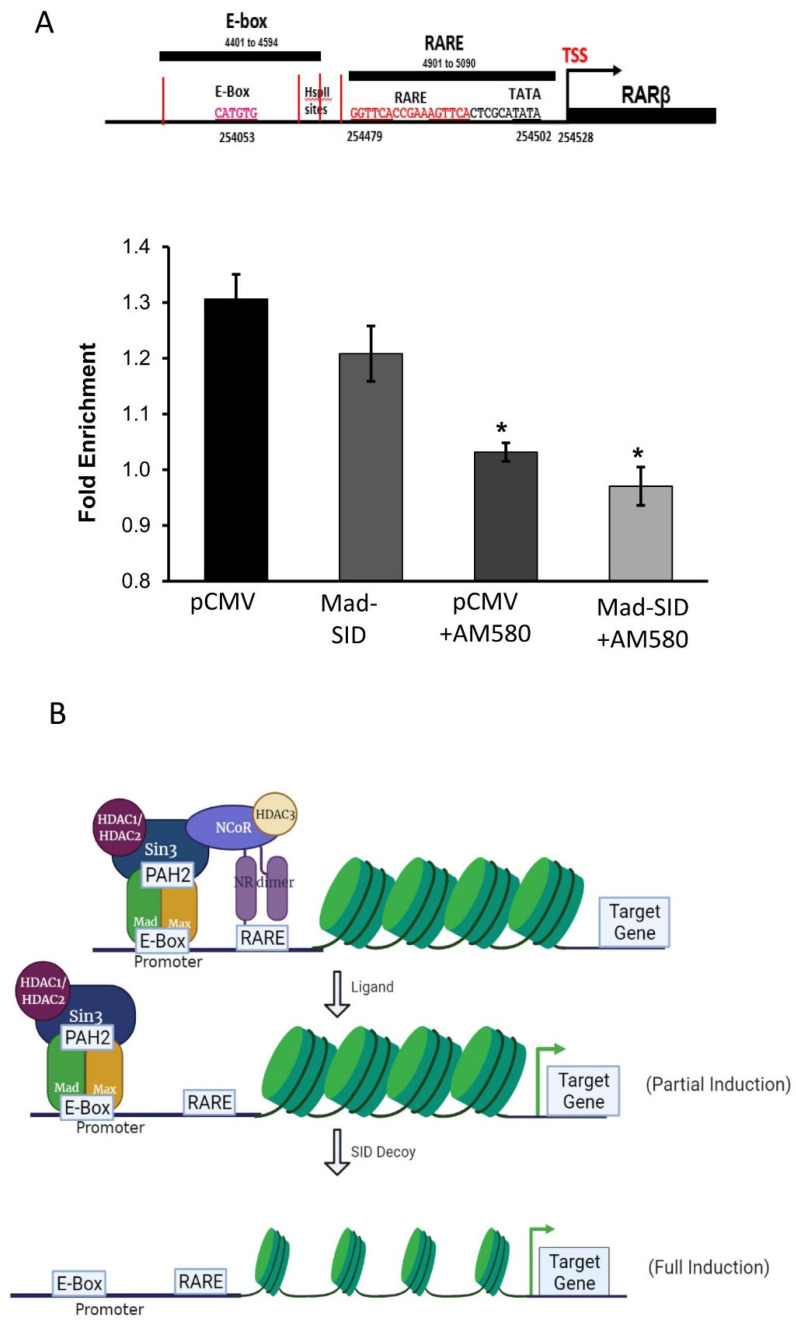
The RARβ expression is controlled by both MAD1 and RARα: Chromatin immunoprecipitation assays performed on AM580 treated pCMV and MAD-SID-overexpressing MDA-MB-231 show a reduction in the enrichment of NCoR at the RARE sequence and (**A**) a reduction of Sin3A and HDAC2 on the E-Box sequence of the RARβ promoter. (**B**) Schematics show the proposed model wherein treatment with either a ligand or SID partially recovers the de-repression of RARβ expression; however, a combination removes the entire complex, leading to complete de-repression of RARβ expression. The data presented are an average change in the fold enrichment obtained from two independent experiments. The statistical analysis was conducted using an unpaired ‘*t*’ test; error bars represent mean ± SD. * *p* < 0.05.

## Data Availability

Not applicable.

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
