# Peer review of "The Sin3A/MAD1 Complex, through Its PAH2 Domain, Acts as a Second Repressor of Retinoic Acid Receptor Beta Expression in Breast Cancer Cells"

_cells, 2022, doi:10.3390/cells11071179_

Round 1
Reviewer 1 Report
MINOR REVISION:
In reading the manuscript I ‘have found some incomplete sentences, minor orthographic and syntax errors.
ABSTRACT
Please correct the following sentence “Sin3A interacts with transcription factors, including MAD1, through their (Sin3 Interaction Domain SID) and chromatin modifying proteins such as histone deacetylases and targets chromatin for specific modifications.” with “Sin3A interacts with transcription factors, including MAD1, through MAD1 Sin3 Interaction Domain (SID) and chromatin modifying proteins such as histone deacetylases and targets chromatin for specific modifications”
Please state in a comprehensive manner that PAH2 is a Sin3A domain.
What does it mean MAD-SID overexpressing cells? Explain better in the abstract.
The following sentence is incomplete: “Chromatin Immunoprecipitation show that MAD-SID…..”.
INTRODUCTION
Please specify estrogen receptor, progesterone receptor etc. in the following sentence.
“Transcriptional activity directed by retinoic X receptor as well as various nuclear receptors including estrogen, progesterone, vitamin D are influenced by epigenetic mechanisms such as DNA methylation at CpG islands and histone modifications”
RESULTS
Please correct the following sentence: “To study the mechanistic basis of increase retinoic acid receptor expression, we generated MDA-MB-231 and MCF-7 cells stably expressing GFP tagged MAD SCR (referred as pCMV) MAD1-SID peptide (Fig.1E) and sorted to achieve a transgene expressing enriched population.”
Please correct the following sentence: “The first clone, clone #30 contained only the RARE which is present at 3bp upstream of the TATA box, clone#24 included both RARE sequence and clone 16/21 contained the entire sequence encompassing the RARE, proximal E-box and distal E-box.” The description of clone 24 is incomplete.
FIG 1
Indicate which is RARa and which is RARb if fig. C and D.
In figure legend expression is written twice in the sentence “Transfected cells were examined for the expression of RARα and RARβ expression through RT-PCR “.
FIG 3
Please describe Fig3C. in the figure legend.
FIG 4
The authors should state which clone they used for the experiment depicted in fig 4D.
DISCUSSION
Please correct the following sentence: “The SMI shows additive effects with RARα ATRA (all- trans retinoic acid) and RARα specific agonists AM80 and AM580.”
MAJOR REVISION
FIG 1
MAD SID overexpression increase RARa mRNA levels in MDAMB231 in fig 1B but not in fig. 1C. Why? The same observation should be done also for RARb comparing Fig 1B and 1D. Moreover, the variation in RARβ mRNA levels upon treatment with AM580 are very small in fig 1D. The reduction in RARβ mRNA levels upon treatment with AM580 in MDAMB231 cells transfected with the pCMV plasmid is unexpected. I would suggest to increase the number of independent experiments performed to support the data depicted in fig 1D.
The data should be corroborated by Western blot analysis of RARa and RARb expression in the cellular models used.
FIG 2
For the indicated IP, a control using IgG should be shown. Lysates should be incubated with the antibody Vs the protein of interest or with the corresponding IgG, as a negative control. The samples should be run in the same gel.
In general, for all the immunoprecipitation shown in fig.2 the levels of immunoprecipitated proteins should be shown and the data of the interacting protein normalized on the amount of immunoprecipitated protein.
FIG 4
Describing fig. 4 the authors state that all the clones “all the clones responded to AM580 treatment.” However, according to fig. 4 there is no difference in the case of clone 8 between luciferase activity on untreated or AM580 treated cells.
Author Response
Reviewer2:
MINOR REVISION:
In reading the manuscript I ‘have found some incomplete sentences, minor orthographic and syntax errors.
Reply: The authors greatly appreciate the effort taken by the reviewer to indicate such errors, it has improved the readability of the manuscript.
ABSTRACT
- Please correct the following sentence “Sin3A interacts with transcription factors, including MAD1, through their (Sin3 Interaction Domain SID) and chromatin modifying proteins such as histone deacetylases and targets chromatin for specific modifications.” with “Sin3A interacts with transcription factors, including MAD1, through MAD1 Sin3 Interaction Domain (SID) and chromatin modifying proteins such as histone deacetylases and targets chromatin for specific modifications”
Reply: The sentence was intended to be as it is written in the manuscript.
- Please state in a comprehensive manner that PAH2 is a Sin3A domain.
Reply: The abstract and introduction have been edited to show that PAH2 is a Sin3A domain.
- What does it mean MAD-SID overexpressing cells? Explain better in the abstract.
Reply: Necessary changes have been made in the abstract.
- The following sentence is incomplete: “Chromatin Immunoprecipitation show that MAD-SID…..”
Reply: Necessary changes have been made in the abstract.
INTRODUCTION
- Please specify estrogen receptor, progesterone receptor etc. in the following sentence “Transcriptional activity directed by retinoic X receptor as well as various nuclear receptors including estrogen, progesterone, vitamin D are influenced by epigenetic mechanisms such as DNA methylation at CpG islands and histone modifications”
Reply: The text has been edited as suggested.
RESULTS
- Please correct the following sentence: “To study the mechanistic basis of increase retinoic acid receptor expression, we generated MDA-MB-231 and MCF-7 cells stably expressing GFP tagged MAD SCR (referred as pCMV) MAD1-SID peptide (Fig.1E) and sorted to achieve a transgene expressing enriched population.”
Reply: The sentence has been corrected.
- Please correct the following sentence: “The first clone, clone #30 contained only the RARE which is present at 3bp upstream of the TATA box, clone#24 included both RARE sequence and clone 16/21 contained the entire sequence encompassing the RARE, proximal E-box and distal E-box.” The description of clone 24 is incomplete.
Reply: Thank you for your comment. Necessary changes have been made.
FIG 1
-Indicate which is RARa and which is RARb if fig. C and D. In figure legend expression is written twice in the sentence “Transfected cells were examined for the expression of RARα and RARβ expression through RT-PCR “.
Reply: Figure legend and figure has been edited.
FIG 3
Please describe Fig3C. in the figure legend.
Reply: The legend has been edited to include “and RARα on RARE sequence”
FIG 4
The authors should state which clone they used for the experiment depicted in fig 4D.
Reply: Although it was mentioned in description that clone containing both RARE and E-box sequence was used for the experiment, for further clarity clone #24 has been included in parenthesis.
DISCUSSION
Please correct the following sentence: “The SMI shows additive effects with RARα ATRA (all- trans retinoic acid) and RARα specific agonists AM80 and AM580.”
Reply: The sentence has been edited.
MAJOR REVISION
FIG 1
MAD SID overexpression increase RARa mRNA levels in MDAMB231 in fig 1B but not in fig. 1C. Why? The same observation should be done also for RARb comparing Fig 1B and 1D. Moreover, the variation in RARβ mRNA levels upon treatment with AM580 are very small in fig 1D. The reduction in RARβ mRNA levels upon treatment with AM580 in MDAMB231 cells transfected with the pCMV plasmid is unexpected. I would suggest to increase the number of independent experiments performed to support the data depicted in fig 1D.
Reply: We have observed that the increase in RARα and RARβ is time and confluence dependent even in MAD-SID overexpressing cells (Supplementary Fig. S1 f and g). The data presented here is of 24h time-point, therefore, we do not observe an increase in the expression of both the receptors, however, we see induction in RARα and RARβ after AM580 treatment in MAD-SID overexpressing cells at this early time-point. The reduction in expression of receptors might be explained by degradation of receptors after a certain amount of time of agonist treatment, which have been reported previously by other researchers and has been discussed in the result section now.
In figure 1C and D, pCMV and MAD-SID overexpressing cells were taken as control to compare the effects of AM580. Therefore, both the groups are taken as 1, due to which both pCMV and MAD-SID overexpressing cells seem to have same expression. The figures have been re-analyzed and presented taking pCMV as control for all the groups. The authors completely agree with the reviewer that more independent replicates such as done in Fig. 1A and B could refine the findings, however, at this stage, the authors cannot conduct further experiments due to logistical issues.
The data should be corroborated by Western blot analysis of RARa and RARb expression in the cellular models used.
Reply: The authors did perform Western blotting to observe changes at protein levels (data presented in supplementary fig. S1 e) however, it was found that levels of RARα and RXRα increased at protein level but only slight (non-significant) increase was observed in RARβ levels. The authors hypothesized that it could be due to another level of post-transcriptional regulation of RARβ expression in TNBC, such as regulation by long non-coding RNAs. It was in the long-term plan to dissect further this layer of regulation, therefore, it didn’t fall in the scope of current manuscript hence the data was not shown.
FIG 2
For the indicated IP, a control using IgG should be shown. Lysates should be incubated with the antibody Vs the protein of interest or with the corresponding IgG, as a negative control. The samples should be run in the same gel. In general, for all the immunoprecipitation shown in fig.2 the levels of immunoprecipitated proteins should be shown and the data of the interacting protein normalized on the amount of immunoprecipitated protein.
Reply: We thank the reviewer for reiterating the protocol which was used by the authors. For all the CoIPs, the IgG control are shown on the left of every blot. Since one IgG control was used for all the three CoIPs i.e. MAD1, RARα and RXRα, and all the fractions were run on the same gel, same IgG has been used for all the blots. Original blots have been submitted for the perusal of the reviewers.
The authors have represented inputs for the CoIPs instead of the immunoprecipitated protein because the band size of heavy chain overlaps RARα (52kD) and RXRα (54kD) proteins. We tried various approaches to get rid of the heavy chains, such as VeriBlot (Abcam, ab131366) without success. Therefore, we had to use inputs as controls and this approach is used by researchers since we normalize the protein concentration before immunoprecipitation.
FIG 4
Describing fig. 4 the authors state that all the clones “all the clones responded to AM580 treatment.” However, according to fig. 4 there is no difference in the case of clone 8 between luciferase activity on untreated or AM580 treated cells.
Reply: The authors have added data from more experiments and have removed clone#8 as it was not further used in the studies, instead another clone (clone#24) which contained both RARE and E-box and showed consistent increase in expression after AM580 treatment or MAD-SID treatment was used for further studies. We have also included positive controls in all the experiments.
Reviewer 2 Report
This is a scientifically sound study and there is little to critique in terms of the experiments and conclusions. To increase the potential applicability of the molecular mechanistic work, the authors should consider how they can test their findings (e.g. the Sin3A/HDAC complex is also mediating the repression of RARβ) using the TCGA patient data. This would broaden the potential impact of the study.
Also, the authors should also review the literature more extensively in terms of the disparate effects of ATRA in breast cancer and known mechanisms for this.
Author Response
Dear Reviewer,
The authors thank you for taking your time to review the manuscript. Below are the responses to your comment:
This is a scientifically sound study and there is little to critique in terms of the experiments and conclusions. To increase the potential applicability of the molecular mechanistic work, the authors should consider how they can test their findings (e.g. the Sin3A/HDAC complex is also mediating the repression of RARβ) using the TCGA patient data. This would broaden the potential impact of the study. Also, the authors should also review the literature more extensively in terms of the disparate effects of ATRA in breast cancer and known mechanisms for this.
Reply: The authors appreciate the encouraging words of the reviewer. We have mentioned in our earlier publications that Sin3A and some of the transcription factors which interact with Sin3A, such as TGIF are overexpressed in breast cancer cells. In the discussion we now briefly address the minimal clinical activity of retinoids such as ATRA in breast cancer treatment.
Reviewer 3 Report
In their previous work, the authors have investigated the impact of peptides that target the interaction between Sin3A, which is a chromatin modulator and one of its interaction partner, the transcription factor MAD1. They have shown that SID peptides, also called SID decoys, induces epigenetic reprogramming and differentiation, inhibits tumorigenesis, and confers retinoid sensitivity in triple negative breast cancer by inducing RARβ expression, a retinoic acid target gene. In the present study, they decided to decipher the mechanism of action of one SID peptide, named MAD-SID, on RARβ expression. To do so, they generated breast cancer cells that stably express MAD-SID and analyzed RARα and RARβ expression by RT-qPCR. They also investigate the effect of AM580, a retinoic acid analog, on RAR expression induced by MAD-SID. They also show that MAD-SID disrupts the interaction between Sin3A and RXR or Sin3A and RARα. By chromatin immunoprecipitation, they show that MAD1 and Sin3A are recruited to the proximal E-box, which is a putative MYC/MAD1 binding sites, on the RARβ gene. Interestingly, MAD-SID reduces MAD1 and Sin3A recruitment. Finally, the authors show that the recruitment of NCoR, a RAR corepressor, to the RARβ promoter is reduced by the combination of MAD-SID and AM580.
Although it is of interest to understand how SID decoys functions to control the expression of a tumor suppressor gene, such as RARβ, the manuscript presents many flaws. There are multiple errors in the presentation of data that detract from the biological significance of the majority of the data presented. Importantly, some important controls are missing.
Major issues:
- Cells that express stably MAD-SID peptide were generated. Is it possible to verify the expression of the peptide? The authors should verify that the stable expression of MAD-SID peptide results in the same cellular effects than that they have previously found, at least on one parameter such as differentiation in 2D cultures. Why do the authors use MCF-7 cells that are a luminal and not a triple negative breast cancer cell line? What is the effect of MAD-SID peptide on cellular parameters in MCF-7 cells?
- The authors did not mention how many times the experiments have been reproduced neither in the figures legends nor in the methods section. Did the authors compare data from replicates from a single experiment or from independent experiments?
- The authors based their conclusions on data that are not significant. In Figure 3, although the p-value is close the usual accepted significance value (0.05), the presence of Sin3A to RARβ promoter is not significant (p=0.051), as the decrease of HDAC1 (p=0.053) in MDA-MD-231 cells transfected with MDA-SID peptide. The significance value for MAD1 recruitment is not given. Is it significant? Same issue for Figure 4D and 5.
- In Figure 2, the authors show cropped gels but they also put cropped gels from different blots side by side which is not acceptable. The authors have to show the entire blots that they used to build their Western blot figure in the Supplemental section. Densitometry on replicates of the protein expression experiments should be performed. Moreover, IgG controls are missing in Figure 2A.
- In the ChIP experiment, AM580 treatment lasts for 24h, as mentioned in the methods section. This is a long exposure to ligand for ChIP. Why do the authors choose this timing? Usually 45 minutes are sufficient to allow the recruitment to nuclear receptors and coregulators to their target gene promoter. Did they authors try shorter ligand exposure? The recruitment of corepressors could be more accurate.
- In Figure 1D, AM580 down-regulates the expression of RARβ in MDA-MB-231 cells. This is not discussed. Is it specific to this cell line? If this effect is significant, then MAD-SID not only sensitizes the cells to AM580 but it completely reverses AM580 effect on RARβ expression. The authors should explain this effect.
- Concerning Figure 4, the authors described the effect of AM580 whereas AM80 is shown in Figure 4C. Which compound has been used? The representation of the reporter genes is different between Figure 4B and 4C, which is confusing. The description of the figure does not suit to the results. For instance, the authors stated that all the clones responded to AM580 treatment whereas Clone#8 does not respond to AM80. There is no positive control treated with AM580 or AM80 in the figure. The authors mentioned “endogenous promoters” which are not cited in the legend. What do the authors mean by “endogenous promoters”? What are full length promoters? Fig. 4E legend mentions synthetic RARE promoters. What are they?
- The authors found that MAD1 and Sin3A bind to the proximal E-box in the RARβ promoter. In the reporter constructs, they should mutate this E-box to confirm its role in the regulation of RARβ promoter regulation.
- In the discussion, the authors conclude that Sin3A-Mad-HDAC complex cooperates with RARα. They should perform luciferase assays with AM580 in combination with MAD-SID to demonstrate the cooperation.
Minor issues
- The authors should review carefully the figure legends. Regarding the statistical analysis, the definition of the p-value is not correctly given in Figure 3, 4 and 5 and looks like a copy and paste from Figure 1 : ** p <0.01 and * p < 0.05 whereas *** are shown on Figure 3 and 4. Moreover, this definition does not suit to the one mentioned in the text for Figure 3.
- Figure S1 has no legend.
- Sin 3A input is missing in Figure 2B. Densitometry is missing on Figure 2C.
- In the legend of Figure 3, the ChIP experiment with RARα is not described.
- In Figure 4, the description of the various reporter genes used in the luciferase assays is confusing. Could the authors use simplest names?
- Figure 4 and its description should be moved before Figure 5. The sentence: “In order to demonstrate that these two sequences (RARE and E-box) have functional relevance for activation of RARβ promoter, we conducted luciferase assays with various re gions of RARβ promoter driving luciferase expression.“ should be moved at the beginning of Figure 4 description.
- Figure 4C instead of Figure 4B in the text, and Figure 4D instead of Figure 4C.
- Fig 4C. The DMSO values do not seem to be at 1.
- In Fig 4D and 4E, the authors should add the untreated control to the graph.
- The description of MAD1 family in the Putative E-box in RARβ promoter binds with MAD1 section should be moved to the introduction.
- In the description of RT-qPCR in the Material and Methods section, the authors mentioned CDH1 and PADI4 genes which are not shown in the results.
- Figure 1 in the text and in the file containing original figures is not the same. Western blots analyses are not in the Figure 1 in the text. Same concern for this Western blot experiment about densitometry on replicate experiments.
- How densitometry of Western blots was performed is not mentioned in the Material and Methods section.
- In the discussion, the authors should refer to the figure they are discussing to help the reader. They do not need to repeat the description of their previous published work at the beginning of the discussion.
Author Response
Dear Reviewer,
The authors thank you for your thorough review of the manuscript, addressal of your critique has improved the readability of the manuscript. Below are our responses to your critique:
Reviewer 3:
Major issues:
- Cells that express stably MAD-SID peptide were generated. Is it possible to verify the expression of the peptide? The authors should verify that the stable expression of MAD-SID peptide results in the same cellular effects than that they have previously found, at least on one parameter such as differentiation in 2D cultures. Why do the authors use MCF-7 cells that are a luminal and not a triple negative breast cancer cell line? What is the effect of MAD-SID peptide on cellular parameters in MCF-7 cells?
Reply: Thank you for the comment. We verified the expression of the peptide (transgene) through RTPCR using forward primer for MAD-SID sequence and reverse primer for GFP sequence, the results have been presented in supplementary figure. For correlating with induction of differentiation or the commitment of cells to a more epithelial phenotype, the authors have also estimated the changes in expression of CDH1 and SOX2, the data have been presented in supplementary figure. In our 2010 paper, we have reported that after overexpression of MAD-SID peptide sequence, we observe an increase in nuclear expression of RARβ in MDA-MB-231 cells. The authors used MCF-7 cells line, being completely aware of the characteristics of the cell line, because we wanted to do a comparative study of expressing MAD-SID in a triple negative and a luminal cell line. The results suggest that MAD-SID expression induces significant changes in luminal cell line.
- The authors did not mention how many times the experiments have been reproduced neither in the figures legends nor in the methods section. Did the authors compare data from replicates from a single experiment or from independent experiments?
Reply: The authors thank the reviewer for pointing out the information which was inadvertently missing. Information about all the replicates has been added in the legend of every figure. The RT-PCR results to confirm induction of RARα and β are representative of four independent replicates, the Co-IP and Western blot data presented was replicated in three independent replicates, Luciferase assay for responsiveness of the constructs to AM580 and MAD-SID and the additive effect experiments are presented from three independent experiments. The ChIP and RT-PCR for induction of RARα and RARβ after AM580 treatment are representative of two independent replicates.
- The authors based their conclusions on data that are not significant. In Figure 3, although the p-value is close the usual accepted significance value (0.05), the presence of Sin3A to RARβ promoter is not significant (p=0.051), as the decrease of HDAC1 (p=0.053) in MDA-MD-231 cells transfected with MDA-SID peptide. The significance value for MAD1 recruitment is not given. Is it significant? Same issue for Figure 4D and 5.
Reply: The authors have edited the results and mentioned ‘borderline significant’ instead of significant for figure 3D and have specified p values on the figure as well.
- In Figure 2, the authors show cropped gels but they also put cropped gels from different blots side by side which is not acceptable. The authors have to show the entire blots that they used to build their Western blot figure in the Supplemental section. Densitometry on replicates of the protein expression experiments should be performed. Moreover, IgG controls are missing in Figure 2A.
Reply: The authors have not cropped random gels and put them together for presentation. The immunoprecipitates of MAD1, RARα and RXRα were run on the same gel and since all the antibodies are anti-mouse, single IgG control was used. Since the CoIP of MAD was not good for presentation while RARα and RXRα IPs were good, we decided to use this blot for showing disruption between RARα and RXRα and Sin3A and use IgG, which was on left side of MAD1. The blots were represented separately for ease of presentation purposes, and thus the same IgG has been cut and represented for the immunoprecipitations which shared same control IgGs. The authors decided not to use the blot for MAD1 immunoprecipitated fraction and performed independent replicates for a better image (shown previously as fig2C, now removed from the manuscript as it represents an independent replicate). However, we have now included the entire blot for presentation and removed the replicate presented earlier as 2C. We have submitted the original blots for perusal of the reviewers.
- In the ChIP experiment, AM580 treatment lasts for 24h, as mentioned in the methods section. This is a long exposure to ligand for ChIP. Why do the authors choose this timing? Usually 45 minutes are sufficient to allow the recruitment to nuclear receptors and coregulators to their target gene promoter. Did they authors try shorter ligand exposure? The recruitment of corepressors could be more accurate.
Reply: We chose ChIP timepoints based on the optimum expression of the clones in luciferase assays and reduction in association of Sin3A and RAR/RXRα interaction in Co-IP assays. The reviewer rightly has suggested that we should have studied shorter time-points to understand the assembly/disassembly of the co-regulators. The results would have been more impressive at lower time points.
- In Figure 1D, AM580 down-regulates the expression of RARβ in MDA-MB-231 cells. This is not discussed. Is it specific to this cell line? If this effect is significant, then MAD-SID not only sensitizes the cells to AM580 but it completely reverses AM580 effect on RARβ expression. The authors should explain this effect.
Reply: As explained to previous reviewer, we observed that the increase in RARα and RARβ is time and confluence dependent even in MAD-SID overexpressing cells (Supplementary Fig. S1 f and g). The data presented here is of 24h time-point, therefore, we do not observe an increase in the expression of both the receptors, however, we see induction in MAD-SID overexpressing cells at this early time-point. Further, The reduction in expression of receptors might be explained by degradation of receptors after a certain amount of time of agonist treatment.
- Concerning Figure 4, the authors described the effect of AM580 whereas AM80 is shown in Figure 4C. Which compound has been used? The representation of the reporter genes is different between Figure 4B and 4C, which is confusing. The description of the figure does not suit to the results. For instance, the authors stated that all the clones responded to AM580 treatment whereas Clone#8 does not respond to AM80. There is no positive control treated with AM580 or AM80 in the figure. The authors mentioned “endogenous promoters” which are not cited in the legend. What do the authors mean by “endogenous promoters”? What are full length promoters? Fig. 4E legend mentions synthetic RARE promoters. What are they?
Reply: The authors apologize for labeling errors. The legend was mislabeled as AM80 instead of AM580, the error has been corrected in the graph. For ease of understanding, the description of luciferase constructs has been rewritten (explaining endogenous promoter) in the methods section and the figure has been modified. The RARE sequence and explanation of synthetic/commercially available kit has been added in the methods section.
- The authors found that MAD1 and Sin3A bind to the proximal E-box in the RARβ promoter. In the reporter constructs, they should mutate this E-box to confirm its role in the regulation of RARβ promoter regulation.
Reply: In order to address the effect of variation in E-box sequence on MAD/c-Myc binding, we have used the distal E-box sequence as negative control in EMSA. It is evident from EMSA results that changes in the E-box sequence severely affects MAD or c-Myc binding to the E-box.
- In the discussion, the authors conclude that Sin3A-Mad-HDAC complex cooperates with RARα. They should perform luciferase assays with AM580 in combination with MAD-SID to demonstrate the cooperation.
Reply: The authors performed more independent replicates for luciferase assays to address this question. The results have been presented in Fig 4C and 4D.
Minor issues
The authors should review carefully the figure legends. Regarding the statistical analysis, the definition of the p-value is not correctly given in Figure 3, 4 and 5 and looks like a copy and paste from Figure 1 : ** p <0.01 and * p < 0.05 whereas *** are shown on Figure 3 and 4. Moreover, this definition does not suit to the one mentioned in the text for Figure 3.
Figure S1 has no legend.
Figure legend is on page 22 of main manuscript.
Sin 3A input is missing in Figure 2B. Densitometry is missing on Figure 2C.
In the legend of Figure 3, the ChIP experiment with RARα is not described.
Answer: Legend for Fig3 has been edited to include RARα. For quantitation and for presentation, we have shown the inputs of proteins which were precipitated.
In Figure 4, the description of the various reporter genes used in the luciferase assays is confusing. Could the authors use simplest names?
Reply: We have re-written the explanations of all the clones and have modified the figure presentation for better understanding of the clone construction.
Figure 4 and its description should be moved before Figure 5. The sentence: “In order to demonstrate that these two sequences (RARE and E-box) have functional relevance for activation of RARβ promoter, we conducted luciferase assays with various re gions of RARβ promoter driving luciferase expression.“ should be moved at the beginning of Figure 4 description.
Reply: Thank you for the suggestion. The sentence has been moved under the results “RARβ promoter responds to AM580 and MAD-SID Treatment”.
Figure 4C instead of Figure 4B in the text, and Figure 4D instead of Figure 4C.
Thank you for noticing the error. The text has been appropriately edited.
Fig 4C. The DMSO values do not seem to be at 1.
Response: The graph has been restructured to normalize the DMSO values.
In Fig 4D and 4E, the authors should add the untreated control to the graph.
Reply: Data from additional experiments has been added to complete figure 4D and 4E (now 4C and 4D)
The description of MAD1 family in the Putative E-box in RARβ promoter binds with MAD1 section should be moved to the introduction.
Reply: The description of MAD1 is discussed in detail in the results section to explain the need for EMSA before the ChIP experiment. Usually researchers find binding sequences from literature and proceed for ChIP, however, in our case for the E-box sequence not much literature was available, thus we have explained this in the results section.
In the description of RT-qPCR in the Material and Methods section, the authors mentioned CDH1 and PADI4 genes which are not shown in the results.
Reply: Thank you for noticing the error, though the expression of the genes was studied, the data in now included only for CDH1 and of SOX2 in addition. Text has been appropriately edited.
Figure 1 in the text and in the file containing original figures is not the same. Western blots analyses are not in the Figure 1 in the text. Same concern for this Western blot experiment about densitometry on replicate experiments.
Reply: Necessary changes have been made both in the text and in the figures.
How densitometry of Western blots was performed is not mentioned in the Material and Methods section.
Reply: The densitometry has been mentioned in Coimmunoprecipitation section.
In the discussion, the authors should refer to the figure they are discussing to help the reader. They do not need to repeat the description of their previous published work at the beginning of the discussion.
Reply: Thank you for the comment, the text in discussion has been appropriately edited to include figures where ever the results have been discussed for better understanding.
Round 2
Reviewer 1 Report
There are still text editing errors, please consider a deeper revision.
Author Response
We thank the reviewer for reviewing our manuscript thoroughly. The manuscript has been re-read and edited wherever necessary.
Reviewer 1:
There are still text editing errors, please consider a deeper revision.
Reply: We thank you for reviewing our manuscript thoroughly. The manuscript has been re-read and edited wherever necessary.
Reviewer 2 Report
ok to accept
Author Response
The authors thank the reviewer for reviewing the manuscript.
Reviewer 2:
ok to accept
Reply: Thank you for accepting our manuscript.
Reviewer 3 Report
The authors have improved their manuscipt. However I still have major concerns.
- The authors now show the expression of MAD-SID by RTPCR. Do they authors perform cell proliferation, migration assays in MCF-7 cells expressing MAD-SID peptide? Performing such assays would improve the reliability in the impact of MAD-SID on cells and not only on gene expression.
- The authors still consider significant the presence of Sin3A to the E-box of RARb promoter although the p-value=0.051. The authors should revise the sentence.
- Concerning western blots, I am sorry but the entire blots are not convincing. The size of the molecular markers should be noted. Moreover, there is no band on the V_231 V IP blots. They authors should show entire inputs blots in the supplementary files. They should also show the B-actin blots used to normalize western blots as mentionned in the M&M sections.
- Minor points:
- english spelling should be verified specially in the new added sentences.
- Legend of Figure 3: the indicated p-values do not correspond to the p-values shown in the figure. ** is missing in the legend.
- Legen of Figure 4 specifies a “E” panel which is not shown.
Author Response
The authors thank the reviewer for reviewing the manuscript.

Round 3
Reviewer 3 Report
I apologize for the misunderstanding. It is a good point that the authors have shown entire gels. It is convincing now that the authors did not performed quantitative comparisons between samples on different gels/blots. However, it is usually of good practice that all unprocessed original images of western blots are published in the Supplementary Information. The size of the markers should be noted on the images. The authors should take care to ensure that the unprocessed scans provided match all the figures. It would seem that unprocessed original images of western blots of Figure 2A and 2C were not submitted. Although the authors mentioned in Material and Methods section that b-actin has been used to normalized expression of RARα and RARβ proteins, it is also of good practice to show that loading controls (e.g. actin) has been run on the same blot. When sample processing controls are run on different gels, they must be identified as such in the figure legend.
Author Response
Comment: I apologize for the misunderstanding. It is a good point that the authors have shown entire gels. It is convincing now that the authors did not performed quantitative comparisons between samples on different gels/blots. However, it is usually of good practice that all unprocessed original images of western blots are published in the Supplementary Information. The size of the markers should be noted on the images. The authors should take care to ensure that the unprocessed scans provided match all the figures. It would seem that unprocessed original images of western blots of Figure 2A and 2C were not submitted. Although the authors mentioned in Material and Methods section that b-actin has been used to normalized expression of RARα and RARβ proteins, it is also of good practice to show that loading controls (e.g. actin) has been run on the same blot. When sample processing controls are run on different gels, they must be identified as such in the figure legend.
Reply: Thank you for clarifying the issue. For β-actin, as a standard lab practice, we always probe the loading control on the same gel. If the size of loading control and protein of interest is close, we first probe for the protein of interest, strip the membrane, block it and re-probe for β-actin. So, all the β-actins have been probed on the same membrane. The request to publish all the unprocessed gels is unusual, which rarely is part of this type of publication and adds little to our data in the manuscript.